# Current Surgical Management Strategies for Colorectal Cancer Liver Metastases

**DOI:** 10.3390/cancers14041063

**Published:** 2022-02-20

**Authors:** Gabriel D. Ivey, Fabian M. Johnston, Nilofer S. Azad, Eric S. Christenson, Kelly J. Lafaro, Christopher R. Shubert

**Affiliations:** 1Department of Surgery, Sidney Kimmel Comprehensive Cancer Center, Johns Hopkins University School of Medicine, Baltimore, MD 21205, USA; igabrie2@jhmi.edu (G.D.I.); fjohnst4@jhmi.edu (F.M.J.); klafaro1@jhmi.edu (K.J.L.); 2Department of Medical Oncology, Sidney Kimmel Comprehensive Cancer Center, Johns Hopkins University School of Medicine, Baltimore, MD 21205, USA; nazad2@jhmi.edu (N.S.A.); echris14@jhmi.edu (E.S.C.)

**Keywords:** colorectal liver metastases, parenchymal-sparing hepatectomy, one- and two-stage hepatectomy, associating liver partition and portal vein ligation for staged hepatectomy, liver transplantation, hepatic arterial infusional chemotherapy, minimally invasive liver resection

## Abstract

**Simple Summary:**

Colorectal cancer is one of the most common cancer diagnoses in the world. At least half of patients diagnosed with colorectal cancer will develop metastatic disease, with most being identified in the liver. Surgical resection of colorectal liver metastases (CRLM) is potentially curative. Surgical resection of CRLM, however, remains underutilized despite the continued expansion of operative strategies available. This is likely due to differing views on resectability. Resectability is a surgical assessment, and the classification of CRLM as unresectable should only be made by an experienced hepatobiliary surgeon. Obtaining a surgical evaluation at the time of liver metastasis discovery may help mitigate the challenge of assessing resectability and the determination of potential operative time windows within current multimodal management strategies. The aim of this review is to help facilitate discussions surrounding resectability as well as the timing and sequencing of both surgical and non-surgical therapies.

**Abstract:**

Colorectal cancer is the third most common cancer diagnosis in the world, and the second most common cause of cancer-related deaths. Despite significant progress in management strategies for colorectal cancer over the last several decades, metastatic disease remains difficult to treat and is often considered incurable. However, for patients with colorectal liver metastases (CRLM), surgical resection offers the best opportunity for survival, can be curative, and remains the gold standard. Unfortunately, surgical treatment options are underutilized. Misperceptions regarding resectable and unresectable CRLM likely play a role in this. The assessment of factors that impact resectability status like medical fitness, technical considerations, and disease biology can be difficult, necessitating careful multidisciplinary input and discussion. The identification of ideal operative time windows that align with the multimodal management of these patients can also be perplexing. For all patients with CRLM it may therefore be advantageous to obtain surgical evaluation at the time of discovering liver metastases to mitigate these challenges and minimize the risk of undertreatment. In this review we summarize current surgical management strategies for CRLM and discuss factors to be considered when determining resectability.

## 1. Introduction

Colorectal cancer is the third most common cancer diagnosis in the world [1], and the second most common cause of cancer-related deaths [2]. Over the last several decades we have seen significant progress in management strategies for colorectal cancer; screening modalities have improved [3], new systemic treatment options have lengthened survival [4], and recent developments in surgical management strategies have expanded resection eligibility criteria [5,6]. Despite this progress, metastatic disease remains the most common cause of death [7]. Metastatic disease, however, is not a contraindication to surgical intervention. For those who develop metastatic disease to the liver, resection of this disease is associated with a 20% cure rate [8], and five-year survival rates can exceed 50% [9].

Unfortunately, surgical management strategies for colorectal liver metastases (CRLM) remain underutilized [10]. Misperceptions, even among surgeons, regarding resectable and unresectable CRLM likely play a role in this [11,12,13]. The assessment of factors that impact resectability status like medical fitness, technical considerations, and disease biology can be difficult. The identification of ideal operative time windows that align with the multimodal management of these patients can also be perplexing. Most patients who have CRLM are seen by medical oncologists for consideration of systemic therapy for management of their metastatic disease, often placing the burden of assessing resectability on medical oncologists alone [10]. To facilitate optimal management all patients with CRLM should have a surgical evaluation at the time of liver metastasis discovery, prior to any treatment decision [14,15], and preferably in the setting of a multidisciplinary conference [16].

The treatment of resectable CRLM traditionally began with chemotherapy and included liver resection as a later option depending on response to systemic treatment. For select patients with resectable CRLM, however, a surgery-first approach may be advantageous [17], and we have recently seen the benefit of perioperative chemotherapy come into question [18,19]. Additionally, there are numerous locoregional therapies for which all patients with CRLM, including those with extrahepatic disease, should be considered [20,21].

While it is difficult to assess a causal relationship between multidisciplinary care and improvement in patient survival [22], it has been suggested that changes in treatment recommendations for patients with gastrointestinal malignancies occur at least one third of the time after multidisciplinary evaluation [23]. For patients with CRLM, it is speculated that at least 20% of patients with isolated liver metastases who have been advised to undergo a chemotherapy-first approach may instead be eligible for upfront surgery [16]. Longitudinal multidisciplinary care of patients with CRLM, including those patients with extrahepatic disease, has the potential to improve overall survival (OS) when compared to systemic therapy alone by increasing resectability and resection rates [24].

In this review we summarize current surgical management strategies for CRLM and discuss factors considered when determining resectability with which all oncology providers should be familiar. Recent trials evaluating the efficacy of current resection techniques are discussed. Systemic treatment strategies are not examined in detail. Ablative locoregional therapies are not reviewed.

## 2. Assessment of Resectability

Classically, patients with CRLM are deemed as having “resectable” or “unresectable” disease with some groups further stratifying patients into a third group called “potentially resectable” or “borderline”. The exact definition of each classification varies across the literature, making comparisons between studies difficult. Despite this, multidisciplinary teams should attempt to define a patient’s resectability for the purposes of treatment planning. Considerations when assessing resectability include an evaluation of disease burden (i.e., size, number, and distribution of CRLM) [25,26,27], impression of disease biology (i.e., rate of disease progression, suspicion of extrahepatic disease, timing of appearance with relationship to the primary colorectal tumor, primary colorectal tumor sidedness, RAS/BRAF mutation status, microsatellite instability [MSI] status) [28,29], and technical aspects like relationship to vascular inflow, outflow, and biliary drainage [11]. In Table 1 we list working definitions for these resectability classifications.

Surgery for CRLM is always performed with curative intent; there is no role for debulking surgery [33]. In this regard, surgical planning revolves around strategizing for the full removal of the tumor while preserving a sufficient remnant of healthy liver tissue (future liver remnant [FLR]) to limit the risk of postoperative liver dysfunction/failure [34]. Resection of disappearing liver metastases should be included in this plan, as more than half of these lesions will recur if left in situ [35,36,37]. Microscopic residual disease is found in up to 90% of specimens harboring radiologically occult liver metastases [38,39,40]. Due to the liver’s enormous regenerative capacity, as much as 80% of a healthy, noncirrhotic liver can be removed [26]. A number of factors, however, like cirrhosis, fibrosis, cholestasis, steatosis, and steatohepatitis can impair the liver’s regenerative capacity [41]. Many commonly used chemotherapeutics, like 5-fluorouracil, irinotecan, and oxaliplatin are known to damage hepatocytes, and their use is often associated with the development of the aforementioned liver injuries [42]. Goal FLR is subsequently dependent on the quality of liver parenchyma. Current guidelines for extended hepatectomies recommend an FLR of ≥20–25% in healthy patients, >30% in patients with chemotherapy-associated liver injury, and >40% in cirrhotics [43].

Preoperative planning therefore requires high-quality liver specific imaging in order to properly assess disease burden and FLR. Computed tomography (CT), magnetic resonance imaging (MRI), and positron emission tomography (PET) are the commonly used modalities for CRLM. Many centers consider modern CT techniques equivalent to MRI, with MRI being helpful for adjudicating indeterminate hepatic lesions [44]. Several recent studies, however, suggest MRI to have higher sensitivity and specificity for CRLM [45,46]. PET is considered a helpful adjunct in select scenarios. Some centers consider PET imaging for further evaluation of indeterminate nodules or soft tissue masses identified on CT or MRI [44]. Others consider PET for patients with an elevated carcinoembryonic antigen (CEA) level, but unremarkable CT or MRI cross-sectional imaging [44]. In one randomized trial that investigated the impact of preoperative PET vs. CT without PET, the use of PET imaging did not frequently change surgical management or have an association with improved survival [47].

Following obtainment of high-quality liver protocol imaging, FLR can be measured using 3-D volumetric software. Three-dimensional volumetric analysis and planning of major liver resections is now standard for some centers as 3-D analysis generates FLR calculations while allowing recognition of and accommodation for intrahepatic anatomical variants, adjustment for tumor volume, and further tailoring for increasingly complex resection strategies.

If the FLR is inadequate to allow for safe resection, but the patient is fit enough for surgery, FLR can be augmented using a variety of techniques (i.e., portal vein embolization, two-stage hepatectomy, associating liver partition and portal vein ligation for staged hepatectomy). Discussions of these strategies are in select sections that follow. Similarly, resectability is not static. For patients with initially unresectable CRLM, chemotherapy has the potential to convert such disease to resectable [48], and can be administered with or without hepatic arterial infusional chemotherapy [49].

## 3. Treatment Sequencing

CRLM represents a spectrum of disease that can present in a variety of manners. Although exact definitions vary, synchronous CRLM are liver metastases that are discovered at or before diagnosis of the primary colorectal tumor [50]. Approximately 14–25% of patients diagnosed with colorectal cancer will be identified with synchronous disease [51,52,53,54]. Colorectal metastases discovered after diagnosis or resection of the primary colorectal tumor are considered metachronous CRLM [50]. About 7–30% of patients with colorectal cancer will develop metachronous disease [51,52,53,54]. These groupings assist in determining the order of treatment strategies and sequencing [55].

### 3.1. Synchronous CRLM

For patients with resectable or borderline resectable synchronous CRLM, three treatment approaches exist: classic, combined, and reversed [56]. Upfront chemotherapy with or without additional radiotherapy for primary rectal tumors is an option for each of these three treatment strategies and should be carefully discussed and decided upon in multidisciplinary fashion. The classic approach sequences surgery for the primary colorectal tumor, followed by adjuvant chemotherapy and then hepatic resection. The combined approach is a combined liver and colorectal resection followed by adjuvant chemotherapy. The reverse approach sequences surgery for the liver metastases, followed by adjuvant chemotherapy and then resection of the primary colorectal tumor.

Determination of an appropriate treatment strategy is dependent on patient fitness, the burden of disease, and institutional experience. For instance, patients with CRLM who have a bleeding or obstructing primary colorectal tumor require more urgent surgical management of their primary colorectal tumor in terms of resection or fecal diversion prior to chemotherapy and liver resection. There are also variants. It is the preference of some surgeons to offer a two-stage hepatectomy where during the first-stage hepatectomy the primary colorectal tumor is also addressed, followed by additional chemotherapy, and ultimately the second-stage hepatectomy. Some institutions prefer to manage both the primary and the CRLM in a single surgery when patient fitness allows.

The treatment paradigm for synchronous CRLM has traditionally always been a chemotherapy-first approach. However, as just described, several treatment sequencing options should be considered for all patients with CRLM when appropriate, including a liver-first approach with or without neoadjuvant chemotherapy. Data supporting this strategy are growing and have demonstrated reasonable short- and long-term outcomes [17,57,58,59]. First proposed in 2006 [58], it is an appealing treatment approach as it focuses attention on the most prognostically important site of disease—the liver. Additionally, should chemotherapy not be administered prior to resection, concerns brought about by chemotherapy-associated liver injury can be avoided (i.e., sinusoidal injury, steatosis), potentially resulting in improved outcomes and the need for a smaller FLR.

Recent evidence suggests that of the surgical sequencing options available for patients with CRLM, a liver-first approach should likely be favored in appropriately selected patients. The findings of a recently published retrospective, propensity-matched study that evaluated 7360 patients with synchronous CRLM revealed that in patients with solitary and multiple unilobar CRLM, survival was similar regardless of treatment strategy (classic, *n* = 4415; combined, *n* = 2393; reversed, *n* = 552). In patients with multiple bilobar metastases, however, a liver-first approach was associated with improved overall survival (3-year OS 69% vs. classic approach 60.4%, *p* = 0.031; vs. combined 54.4%, *p* = < 0.001) [17]. Compared with the other groups, the liver-first cohort had more rectal tumors (58% vs. 31.2%), a higher tumor burden, and were more likely to receive neoadjuvant chemotherapy (75%) [17]. These findings highlight the nuances that must be considered when assessing treatment options for patients with CRLM and the benefit of early surgeon involvement.

### 3.2. Metachronous CRLM

For patients with resectable or borderline resectable metachronous disease, treatment approaches are much simpler compared to synchronous disease: surgery alone or surgery with perioperative chemotherapy [60]. There are numerous variants within the perioperative chemotherapy approach. It is the preference of some providers to administer chemotherapy before surgical intervention, after surgical intervention, or both before and after.

### 3.3. Systemic Chemotherapy Sequencing

Both synchronous and metachronous CRLM have traditionally been treated with perioperative chemotherapy, but the benefits of this management strategy have recently come under scrutiny. Results from two randomized controlled trials that evaluated the effectiveness of surgery alone versus surgery with perioperative chemotherapy in patients with synchronous and metachronous CRLM suggest that while perioperative chemotherapy may improve progression-free survival (PFS) or disease-free survival (DFS), there does not seem to be a benefit in OS [18,19]. These findings are also clinically significant as treatment with FOLFOX (folinic acid, fluorouracil, and oxaliplatin), the chemotherapy strategy used in each of these studies, places patients at risk for chemotherapy-related toxicities and adverse events. One well-recognized phenomenon, for instance, is that of chemotherapy-induced peripheral neuropathy, which can substantially affect quality of life [61].

In the European Organisation for Research and Treatment of Cancer (EORTC) 40983 trial, the authors investigated the impact of FOLFOX on PFS and OS in patients with resectable CRLM and up to four liver metastases. A total of 364 patients with resectable CLRM were randomized to either perioperative FOLFOX4 or surgery alone. Just over 50% of enrolled patients had primary colon cancer, about 60% had metachronous disease, and a majority had either one or two liver metastases. Patients in the perioperative chemotherapy arm experienced improved three-year PFS (36.2 percent vs. 28.1 percent, *p* = 0.041) at the expense of increased postoperative complications (25% vs. 16%, *p* = 0.04) [60]. No difference in OS was observed between the two cohorts (61.3 vs. 54.3 months), although the trial was not powered upfront to detect a difference in OS [18].

In the Japan Clinical Oncology Group Study (JCOG0603) trial, the authors investigated the impact of FOLFOX on DFS and OS in patients with any number of CRLM. A total of 300 patients were randomly assigned to either hepatectomy alone or hepatectomy followed by mFOLFOX6. The trial was notably terminated early at the third interim analysis per protocol because DFS was significantly longer in patients treated with hepatectomy followed by chemotherapy (49.8% vs. 38.7% at 5 years, *p* = 0.006) [19]. Just over 75% of enrolled patients had primary colon cancer, about 45% had metachronous disease, and approximately 90% had 1–3 liver metastases [19]. No difference in postoperative complications or 5-year OS (hepatectomy alone 83.1% vs. 71.2%, *p* = 0.42) was observed between the two cohorts [19].

Taken together, these studies highlight the wealth of information available on current treatment strategies for CRLM that we must carefully discuss with our patients when determining a treatment approach. Benefits in DFS and PFS for patients are not insignificant, but they can easily be misperceived as benefits in overall survival [62]. The data on management strategies surrounding metachronous and synchronous CRLM are vast and there is no one-size-fits-all when it comes treatment strategy, including the selection of a surgical approach. Discussions with patients regarding potential surgical intervention must also be carefully framed and communicated. While surgical interventions have the potential to provide benefit to a select cohort, they are not without morbidity and mortality risks that must be carefully weighed.

### 3.4. Considerations When Sequencing Chemotherapy before Hepatectomy

Should chemotherapy be administered prior to hepatectomy, careful consideration should be given to the duration of chemotherapy administration and the potential liver-specific injuries that can result.

It has been suggested that for patients with resectable CRLM, resection should be considered after 2–4 months of chemotherapy so long as there is no evidence of disease progression [63]. In one single institution study, it was observed that for patients treated with chemotherapy alone or HAIC combined with systemic chemotherapy, additional tumor response was not appreciated beyond 4 months in either group [63]. Similar findings were observed in another single-institutional study that examined 407 patients with CRLM who underwent hepatectomy after neoadjuvant chemotherapy [64]. The authors of this study observed the optimal duration of neoadjuvant chemotherapy to be 5 cycles or less, and that treatment beyond 5 cycles was not associated with significant differences in R0 resection rates, pathological response, or postoperative complications. Receipt of greater than 5 cycles of neoadjuvant chemotherapy was also observed on multivariate analysis to be associated with reduced PFS (HR = 1.808, 95% CI 1.205–2.712, *p* = 0.004) and OS (HR 1.723, 95% CI 1.041–2.851, *p* = 0.034).

In addition to duration of chemotherapy administered, regimen is also of significance when assessing the sequelae of chemotherapy. Liver injury is regimen-specific, with oxaliplatin-based regimens often associated with sinusoidal injury and irinotecan-based regimens often associated with steatohepatitis [65]. These impairments have the potential to impact surgical morbidity and mortality [66,67]. In the EORTC 40983 trial, patients in the perioperative chemotherapy arm experienced more postoperative complications than those in the surgery-alone arm (25% vs. 16%, *p* = 0.04) [60]. No differences in mortality were observed. Interestingly, in the JCOG0603 trial, the incidence of perioperative complications between arms was similar [19]. However, one patient died of unknown cause after 3 courses of neoadjuvant mFOLFOX6.

## 4. Surgical Management Strategies

Once an assessment of medical fitness has been established, and a treatment sequencing approach selected, there are several hepatic resection operations for CRLM (Figure 1) that can be considered. This recently expanded selection of surgical approaches is the direct result of improved understanding of segmental anatomy [68], the importance of inflow occlusion (i.e., Pringle maneuver) [69], and low central venous pressure anesthesia [70]. Operations offered to patients must be individualized with the goal of performing a complete resection of all radiographically visible disease (including disappearing liver metastases given the risk of recurrence) [36,38], maximizing FLR, and preserving vascular inflow, outflow, and biliary drainage.

### 4.1. Parenchymal-Sparing Hepatectomy

Parenchymal-sparing hepatectomy, also called non-anatomic liver resection(s), relies on the principal of preserving non-tumorous liver tissue. Tumors are resected with as little normal hepatic parenchyma as possible without the need for pre-operative techniques to induce liver hypertrophy, like portal vein embolization (PVE), portal vein ligation (PVL), or liver venous deprivation (LVD). Indications for this technique include unilobar and bilobar disease.

This technique is considered oncologically equivalent to anatomic or major liver resections, and is associated with lower postoperative morbidity and shorter hospital stays [71]. Use of this technique does not increase the risk of recurrence in the liver remnant, and in fact allows for easier salvage therapy in case of liver recurrence [72]. Because of this it is considered the preferred method for the treatment of resectable CRLM, if allowed by tumor size and location [71].

Some centers employ extreme parenchymal-sparing techniques as repeat hepatectomy for recurrent CRLM can be safely performed in select patients [73]. Recurrence in the residual liver occurs approximately 33% of the time [18], and if maximal normal parenchyma was not spared during the initial hepatectomy, the odds of successfully performing a repeat hepatectomy are significantly reduced. Extreme parenchymal-sparing techniques employ ultrasound for performance of vessel-guided hepatectomy. In select centers, outcomes are comparable to that of alternative surgical management strategies like two-stage hepatectomy (TSH) and associating liver partition and portal vein ligation for staged hepatectomy (ALPPS) for patients with multiple, bilobar, deeply located CRLM [74,75].

Parenchymal-sparing hepatectomy can be performed in the classic, combined, or reversed approaches for synchronous disease or alone (with or without perioperative systemic therapy) for metachronous disease.

### 4.2. One-Stage Hepatectomy with or without PVE/HVE

One-stage hepatectomy is a liver resection that is performed with or without preoperative techniques to induce hypertrophy of the FLR. Should pre-operative hypertrophy be required due to a small FLR (i.e., <30%), PVE with or without hepatic vein embolization (HVE) can be performed. Following sufficient hypertrophy, hepatectomy can then be performed. Disease in patients undergoing one-stage hepatectomy can be multifocal, and this strategy can be combined with parenchymal-sparing hepatectomy of the FLR if disease is bilobar. Patients with bilobar disease who require hypertrophy of the FLR, however, are typically not managed via one-stage hepatectomy.

Historically, one-stage hepatectomy for patients with CRLM was considered superior to parenchymal-sparing hepatectomy [76], but several studies have now shown parenchymal-sparing techniques to be oncologically equivalent and associated with lower morbidity and mortality [71,77]. One-stage hepatectomy strategies are therefore preferred only in select circumstances and usually related to anatomy and/or tumor burden [44].

One-stage hepatectomy can be performed in the classic, combined, or reversed approaches for synchronous disease or alone (with or without perioperative systemic therapy) for metachronous disease.

### 4.3. Two-Stage Hepatectomy

A two-stage hepatectomy is a sequential liver resection where during the first operation the planned FLR is surgically cleared of disease. Following tumor clearance from the FLR, the contralateral portal vein is either ligated or embolized to promote hypertrophy of the FLR. Once sufficient hypertrophy has been achieved, the second liver resection removes the remaining diseased liver.

This strategy takes advantage of the liver’s regenerative capacity allowing patients with significant bilobar disease a chance at cure. Use of this technique has grown exponentially since 2000, when the first series of two-stage hepatectomy in patients with unresectable bilateral CRLM was published [78]. Results from the largest two-stage hepatectomy series in the US, published in 2021, demonstrate it to be a safe and feasible procedure for patients with bilobar disease [79]. Among the 196 patients who underwent two-stage hepatectomy in this series for a median number of 7 tumors, median OS was 50 months [79]. PVE was performed in 128 (65.3%) patients and a majority of patients received chemotherapy prior to the first stage than after the second stage (92% vs. 60%, *p* = 0.308) [79].

Two-stage hepatectomy is more commonly performed in the combined or reversed approaches for synchronous disease or alone (with or without perioperative systemic therapy) for metachronous disease. When combined resection for synchronous disease is chosen, the resection of the primary colorectal cancer is typically performed at the time of first-stage liver resection.

### 4.4. Associating Liver Partition and Portal Vein Ligation for Staged Hepatectomy (ALPPS)

An associating liver partition and portal vein ligation for staged hepatectomy (ALPPS) procedure is a two-stage hepatectomy variant. Like the conventional two-stage hepatectomy, the goal of the first operation is to clear the planned FLR of disease. Following this, the contralateral portal vein is ligated and the right and left hemilivers are divided without disturbance to the remaining vascular and biliary pedicles. Once sufficient hypertrophy is achieved, which typically occurs at a faster rate compared to PVE, the second liver resection removes the remaining diseased liver.

When performed at experienced centers in well-selected patients, there are data that suggest it to be superior to two-stage hepatectomy [5,80]. In a cohort of 100 patients with CRLM and standardized FLR (sFLR) <30% who were randomized to either ALPPS or two-stage hepatectomy, patients randomized to ALPPS had a higher resection rate (92% vs. 80%, *p* = 0.091), and improved OS (46 vs. 26 months, *p* = 0.028) without differences in complications (43% vs. 43%, *p* = 0.99), 90-day mortality (8.3% vs. 6.1%, *p* = 0.68), or R0 resection rates (77% vs. 57%, *p* = 0.11) [5,80].

ALPPS is most commonly performed in the combined approach for synchronous disease or alone (with or without perioperative systemic therapy) for metachronous disease. When combined resection for synchronous disease is chosen, the resection of the primary colorectal cancer is typically performed at the time of first-stage liver resection.

### 4.5. Liver Transplantation

Orthotopic liver transplantation (OLT) removes the entire diseased liver and replaces it with a normal liver (partial or whole) from a deceased or living donor. It is a very rare procedure performed for CRLM, but mounting evidence mainly from European centers with large donor pools [81] suggests that it may provide survival benefit for select patients [6]. The trialing of transplantation as a potential strategy for patients with CRLM has been the result of continued recurrence risk following curative intent liver resection. As many as two-thirds of patients undergoing curative intent liver resection will experience disease recurrence, half of which occurs in the remnant liver [18].

The tractability of this approach outside of select institutions with large donor pools and significant experience remains to be seen. The global transplant community continues to face a shortage of organs [82], and current indications for OLT for patients with CRLM are not well-defined [81]. Experience with OLT at Oslo University Hospital in Norway between 2006 and 2019 suggests that transplant for patients with unresectable colorectal metastases and left-sided primary tumors have improved 5-year OS compared to PVE and liver resection (45.3% vs. 12.5%, *p* = 0.04) [6].

Other important poor prognostic features that should be considered when evaluating transplant candidacy in addition to right-sided disease are BRAF and RAS mutational status, progression on chemotherapy, and N2 nodal status of the primary. Transplantation is not recommended for patients with BRAF V600E mutations but can be considered for patients with RAS mutations despite their poor prognostic association [83]. Transplantation is not recommended if there is evidence of radiological or biochemical progression of disease during the 6 months of required bridging therapy [83]. Nodal disease of N2 of the primary tumor is a relative exclusion criterion [83].

### 4.6. Hepatic Arterial Infusional Chemotherapy

Hepatic arterial infusional chemotherapy (HAIC) is an additional surgical and chemotherapeutic management strategy that should be considered in patients with CRLM (Figure 2). Utilization of this strategy requires subcutaneous placement of a hepatic arterial infusional pump (HAIP)—a metallic device about the size of a hockey puck. Connected to this device is a catheter, which during placement is inserted into the gastroduodenal artery, allowing direct arterial access for the administration of agents with high first-pass hepatic extraction (i.e., floxuridine [FUDR]) limiting systemic toxic side effects, and allowing for the concomitant administration of systemic chemotherapy [84,85]. While it is not a liver resection strategy, it can be placed following hepatectomy for adjuvant liver-directed chemotherapy [86,87]. Traditionally, however, it has often been used as a strategy to convert patients with initially unresectable CRLM to resectable [88].

The pairing of HAIC with systemic chemotherapy has the potential to significantly augment response rates for patients with initially unresectable CRLM. Objective response rates for this cohort following the administration of systemic therapy alone are around 64% (range, 43–79%), generating resection rates of about 23% [89]. When systemic therapy is administered concomitantly with HAIC, objective responses as high as 85% have been observed in patients who have previously received chemotherapy [90], and as high as 100% in chemotherapy naïve patients [91]. Rates of conversion to resectability following this combination have been reported as high as 52% [92].

The expanded use of this modality has been hindered by the specialized expertise needed for pump placement, use, and maintenance. Patients with HAIPs must return for follow-up visits every 2 weeks in order to maintain the reservoir system, treatment facilities must be able to manage complications that can occur following placement (i.e., port migration, catheter dislocation, arterial occlusion, etc.), and chemotherapy-administering teams must know how to properly access the pump, troubleshoot malfunctions, and manage toxicities associated with combined HAIC and systemic chemotherapy [86,93,94].

### 4.7. Repeat Hepatectomy for Recurrence

Hepatic recurrence following curative intent hepatectomy for CRLM occurs approximately 33% of the time [18]. For select patients, repeat hepatectomy is associated with improved OS if there is a sufficient remnant of healthy liver tissue [95,96]. Ahmed at al. examined 274 consecutive patients who underwent resection of CRLM, of which 64 developed metastases confined to the liver. Five-year OS was significantly higher for the 19 patients who underwent repeat hepatectomy compared to the 45 patients who did not (73% vs. 43%, *p* = 0.03) [95]. Factors predictive of worse OS identified on multivariate analysis were time interval less than 1 year between resection of the primary colorectal tumor and liver resection (*p* = 0.001), more than 3 CRLM (*p* = 0.001), no repeat hepatectomy (*p* = 0.01), and lymph node-positive colorectal cancer (*p* = 0.02).

In another study by Battula at al. that examined 969 patients who underwent hepatic resection for CRLM at a single institution over a 13-year period, repeat hepatectomy was also observed to be associated with improved long-term survival in a select cohort [96]. For the 53 patients who were identified as having undergone a repeat hepatectomy, 5-year OS was observed to be 52%. Factors predictive of worse OS identified on Cox regression analysis were R1 resection at first hepatectomy (*p* = 0.002), short time interval between the first and second hepatectomies (*p* = 0.02), and the presence of extrahepatic disease (*p* = 0.02). 

## 5. Minimally Invasive Liver Resection

In experienced centers, laparoscopic and robotic surgery for CRLM is safe and capable of providing equivalent oncologic outcomes compared to open approaches [97,98,99]. The option of a minimally invasive approach is a possibility for most open cases, albeit with greater difficulty depending on the size and location of the CRLM [99]. Like other minimally invasive operations that have been compared to their open counterparts, minimally invasive liver resection is also often associated with lower blood loss and shorter hospital stay [100]. Combined minimally invasive colorectal and liver excision surgery is also feasible and safe but requires an expert surgical team in both minimally invasive colorectal and liver surgery [101].

## 6. Conclusions

Surgical resection of CRLM remains the gold standard and the best opportunity for long-term survival. For a select cohort of patients with CRLM, surgical resection can be curative. Refinements in the understanding of surgical anatomy along with surgical technique have resulted in an expanded assortment of available surgical approaches and have expanded what is considered resectable.

Surgical management strategies, however, remain underutilized despite this progress, likely the result of misperceptions surrounding resectability. Size, number, and distribution of CRLM can be prognostic, along with the presence of extrahepatic disease, but carefully selected and sequenced multimodal treatment strategies can result in improved survival, including surgery-first approaches. Early, upfront, and prospective involvement of a surgeon with knowledge and experience in liver surgery should be considered following the diagnosis of CRLM in order to minimize the risk of undertreatment.

## Figures and Tables

**Figure 1 cancers-14-01063-f001:**
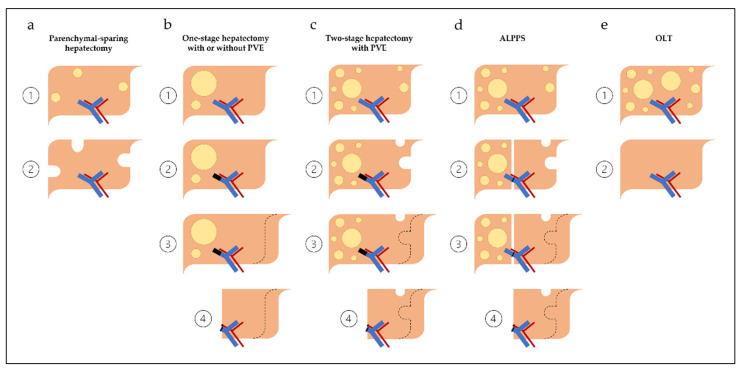
Surgical strategies for colorectal liver metastases. (**a**) Parenchymal-sparing hepatectomy. (**b**) One-stage hepatectomy with or without PVE. (**c**) Two-stage hepatectomy with PVE. (**d**) Associating liver partition and portal vein ligation for staged hepatectomy (ALPPS). (**e**) Orthotopic liver transplantation (OLT). Dashed lines illustrate the future liver remnant prior to augmentation (i.e., PVE; portal vein ligation during ALPPS).

**Figure 2 cancers-14-01063-f002:**
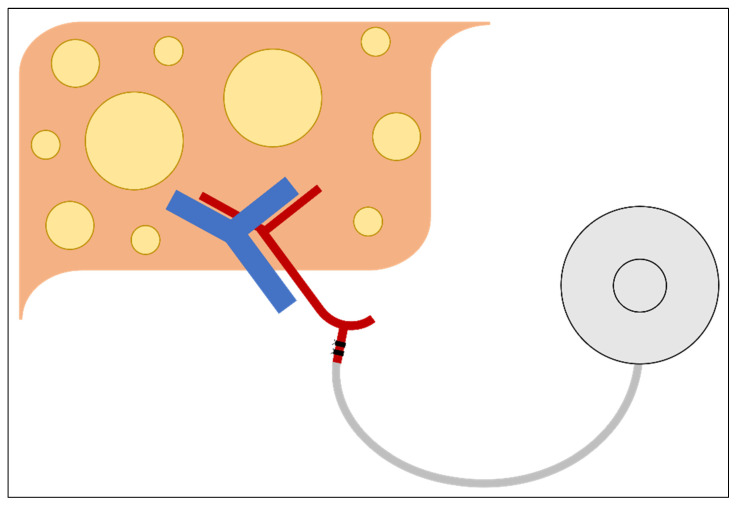
Hepatic arterial infusional pump.

**Table 1 cancers-14-01063-t001:** Definitions of common colorectal liver metastases resectability classifications.

Resectability Classification	Definition
Resectable	The CRLM can be completely resected, two adjacent liver segments can be spared, adequate vascular inflow and outflow and biliary drainage can be preserved, and the volume of the future liver remnant will be adequate (i.e., at least 20% of the total estimated liver volume) [30].
Borderline	The CRLM can potentially be completely resected, but there may be technical (i.e., odds of achieving an R0 resection are reduced) and/or biological challenges (i.e., numerous liver metastases, evidence of disease progression, possible extrahepatic disease) [31].
Unresectable	The CRLM cannot be resected due to burden of disease (i.e., greater than 70% of the liver involved or more than six segments, invasion of both portal veins or all hepatic veins) [32].

CRLM, colorectal liver metastases.

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
