# Peer review of "Current Surgical Management Strategies for Colorectal Cancer Liver Metastases"

_cancers, 2022, doi:10.3390/cancers14041063_

Round 1

Reviewer 1 Report

Dear Authors,

I read your article with great interest. However, I do recommend some minor revisions:

  1. Firstly, the tone and the content of the article regarding perioperative management is very surgery oriented and while there is debate regarding the systemic therapy part, surgery first is not always the ‘best route’ as pointed in several places where there’s equipoise.
  2. I also saw no discussion regarding choosing which type of chemotherapy backbone. I saw no discussion regarding doublet or triplet chemotherapy. ‘Conversion’ therapy is also an important consideration in patients with CRLM who may be deemed unresectable at front but then may become resectable later.
  3. I did not see any discussion or articles pertaining to the biology (RAS/BRAF) and CRLM. There is growing literature ad large studies pertaining to this. A review article without any discussion on biology and systemic therapy would be incomplete.
  4. Furthermore, it is important to point out nuances regarding biologics. Like for example resectable CRLM, data on use of anti-EGFR leading to worse outcomes. These are practical considerations to cite.
  5. Along the discussion on biology, can’t have a complete discussion regarding what to do with the MSI-High tumors. There are series to report pathologic complete responses in these setting. Should these patients be getting surgery first? Or assessment of response on immunotherapy since it is approved now after the Keynote 177 data for advanced or metastatic tumors.
  6. Did not see details on future directions, e.g., trials in this space factoring in ctDNA.
  7. Page 3: the statement regarding all systemic therapies causing liver steatohepatitis is incorrect. As noted, later page 6, different drugs cause different issues. Anti-EGFR don’t cause any liver damage. Bevacizumab in fact protects and there is a meta-analysis showing protection to hepatocytes. That paragraph needs to be corrected.

While the surgical piece is well written, the medical oncology and biology piece warrants more discussion including upcoming trials and conversion therapy.

Overall, the topic and manuscript does warrant publication but needs the following additions/edits.

Good Wishes.

Author Response

Dear Reviewer 1,

Thank you so much for your review. Appreciate your feedback. We hope that the issues you brought up are now addressed.

1. Firstly, the tone and the content of the article regarding perioperative management is very surgery oriented and while there is debate regarding the systemic therapy part, surgery first is not always the ‘best route’ as pointed in several places where there’s equipoise.

We could not agree with you more. We refrained from using language like “surgery-first should be considered standard of care” and instead opted for statements like “a surgery-first approach may be advantageous.” (Section 1, Paragraph 3)

2. I also saw no discussion regarding choosing which type of chemotherapy backbone. I saw no discussion regarding doublet or triplet chemotherapy. ‘Conversion’ therapy is also an important consideration in patients with CRLM who may be deemed unresectable at front but then may become resectable later.

We opted not to discuss chemotherapy regimens given the many options available. We do highlight, however, that chemotherapy has the potential to convert unresectable CRLM to resectable. (Section 2, Paragraph 5)

3. I did not see any discussion or articles pertaining to the biology (RAS/BRAF) and CRLM. There is growing literature ad large studies pertaining to this. A review article without any discussion on biology and systemic therapy would be incomplete.

As you pointed out, we did not highlight RAS/BRAF mutational status when noting that impression of disease biology should be a consideration when assessing resectability. We have since corrected this. (Section 2, Paragraph 1)

We have also added the following citation:

Dijkstra, M., S. Nieuwenhuizen, R.S. Puijk, F.E.F. Timmer, B. Geboers, E.A.C. Schouten, J. Opperman, H.J. Scheffer, J.J.J. de Vries, K.S. Versteeg, et al., Primary Tumor Sidedness, RAS and BRAF Mutations and MSI Status as Prognostic Factors in Patients with Colorectal Liver Metastases Treated with Surgery and Thermal Ablation: Results from the Amsterdam Colorectal Liver Met Registry (AmCORE). Biomedicines, 2021. 9(8).

Mutation status for RAS and BRAF had otherwise only been mentioned in the last paragraph of subsection 4.5.

4. Furthermore, it is important to point out nuances regarding biologics. Like for example resectable CRLM, data on use of anti-EGFR leading to worse outcomes. These are practical considerations to cite.

We recognize there are significant nuances regarding biologics and other treatment strategies. We opted not to discuss these regimens given the need for lengthy discussion.

5. Along the discussion on biology, can’t have a complete discussion regarding what to do with the MSI-High tumors. There are series to report pathologic complete responses in these setting. Should these patients be getting surgery first? Or assessment of response on immunotherapy since it is approved now after the Keynote 177 data for advanced or metastatic tumors.

As you pointed out, we did not highlight the significance of MSI status when noting that impression of disease biology should be a consideration when assessing resectability. We have since corrected this. (Section 2, Paragraph 1)

We have also added the following citation:

Dijkstra, M., S. Nieuwenhuizen, R.S. Puijk, F.E.F. Timmer, B. Geboers, E.A.C. Schouten, J. Opperman, H.J. Scheffer, J.J.J. de Vries, K.S. Versteeg, et al., Primary Tumor Sidedness, RAS and BRAF Mutations and MSI Status as Prognostic Factors in Patients with Colorectal Liver Metastases Treated with Surgery and Thermal Ablation: Results from the Amsterdam Colorectal Liver Met Registry (AmCORE). Biomedicines, 2021. 9(8).

6. Did not see details on future directions, e.g., trials in this space factoring in ctDNA.

We recognize this is an important area of study. However, we opted not to discuss considerations like ctDNA, as it remains unclear how it will impact surgical management strategies for CRLM.

  1. Page 3: the statement regarding all systemic therapies causing liver steatohepatitis is incorrect. As noted, later page 6, different drugs cause different issues. Anti-EGFR don’t cause any liver damage. Bevacizumab in fact protects and there is a meta-analysis showing protection to hepatocytes. That paragraph needs to be corrected.

This paragraph has been corrected.

Reviewer 2 Report

Good review of current management of CRCLM.

Author Response

Thank you so much for your review.

Reviewer 3 Report

Thank you for your well-designed and complete review paper. I like it. The authors completely covered the aim of the paper discussing the CRLM resectability criteria,  treatment sequence as well as surgical and non-surgical treatment strategies.

Author Response

Thank you so much for your review.